# Multiple Physical Symptoms Are Useful to Identify High Risk Individuals for Burnout: A Study on Faculties and Hospital Workers in Japan

**DOI:** 10.3390/ijerph18063246

**Published:** 2021-03-21

**Authors:** Yuki Chatani, Kyoko Nomura, Haruko Hiraike, Akiko Tsuchiya, Hiroko Okinaga

**Affiliations:** 1Department of Anesthesiology, National Hospital Organization Saitama Hospital, Saitama 351-0102, Japan; iprofm_29682@yahoo.co.jp; 2Department of Public Health, Akita University Graduate School of Medicine, Akita 010-8543, Japan; 3Department of Obstetric and Gynecology, Teikyo University Itabashi Hospital, Tokyo 173-0003, Japan; haruko.hiraike@gmail.com; 4Division of Nursing, Teikyo University Itabashi Hospital, Tokyo 173-0003, Japan; a-tutiya@med.teikyo-u.ac.jp; 5Support Center for Women Physicians and Researchers, Teikyo University, Tokyo 173-0003, Japan; hiokinaga@gmail.com

**Keywords:** physical symptoms, burnout, Copenhagen Burnout Inventory, hospital workers, faculties, work–family conflict

## Abstract

Healthcare workers have a high risk of burnout. This study aimed to investigate if the numbers of physical symptoms are associated with burnout among healthcare workers. We conducted a cross-sectional survey at a large university in Tokyo, Japan, in 2016. Participants were 1080: 525 faculties and 555 hospital workers. We investigated 16 physical symptoms perceived more than once per week and examined the association between the number of physical symptoms and Copenhagen Burnout Inventory (CBI); work-related (WBO), personal (PBO), and client-related (CBO) burnout. All CBI scores were higher among hospital workers than among faculties: WBO (43 vs. 29), PBO (50 vs. 33), CBO (33 vs. 29). Moreover, the higher the number of physical symptoms perceived, the higher the degree of burnout scores became (trend *p*-values < 0.001), except for CBO among faculties. Job strain (all except for CBO among hospital workers) and work–family conflict were associated with an increased risk of burnout. Being married (WBO and CBO among faculties), having a child (except for PBO and CBO among faculties), and job support (faculty and hospital workers with WBO and faculties with PBO) were associated with a decreased risk of burnout. Multiple physical symptoms might be useful for identifying high risk individuals for burnout.

## 1. Introduction

During their daily clinical work, healthcare professionals sometimes encounter patients who report physical complaints—such as fatigue, backache, headache, and stomachache—with no apparent cause. Usually, they claim various kinds of physical symptoms but healthcare professionals often cannot find any underlying diseases. In such cases, we had the hypothesis that it can be due to mental health malfunction. There are a few studies to support our theory and one previous study suggested that patients with multiple unspecified physical complaints often have mental health problems [1,2]. Among previous studies that investigated the association between physical symptoms and mental health, one study on Latino and Asian Americans found that people with more physical symptoms with no apparent cause often seek mental health service [3].

Another study concurred with this theory and demonstrated that unspecified eye symptoms—such as blurred vision and trouble seeing without a specific diagnosis—are associated with worse self-perceived mental health [4]. Burnout is a mental health issue characterized by a state of emotional, physical, and mental exhaustion [5]. In this study, we focused on burnout among faculties and hospital workers; these individuals are particularly vulnerable to burnout, due to the daily stress of managing patients and coping with their lives and deaths, dealing with patients’ families, and often working in conditions, including frequent night shifts and overtime. More recently, the situation may be even worse because of the COVID-19 pandemic, which causes faculties and hospital workers—including doctors and nurses—to experience intense psychological pressure [6]. Ignoring faculty and hospital workers’ possible burnout may lead to them developing adverse health outcomes such as depression, which could, in turn, influence the healthcare system. In this regard, early intervention to address burnout benefits both workers and organizations, as high turnover rates—due to burnout—engender an additional financial burden to train new employees [7] as well as more extra personnel and overworked employees who remain in the workplace [8]. Previous studies emphasized burnout as highly prevalent among doctors [9,10], nurses [11,12], and medical faculties [13]; however, there are very few informative and practical studies that address the early indicators of burnout among healthcare professionals. We previously demonstrated that, among healthy white-collar workers, the number of self-reported physical symptoms is significantly associated with job stress [14], as measured by the Job Content Questionnaire (JCQ) [15]. In this study, we applied this concept to medical professionals to examine whether the number of physical symptoms could be used as an early indicator of burnout. It is known from previous studies that medical professionals usually hesitate to seek help at the workplace because they believe that disclosure of mental illness will harm their professional career [16]. If multiple physical symptoms are useful indicators of burnout, the self-reporting of multiple physical symptoms may be a good alternative to direct intervention for burnout in this population. This study therefore aimed to investigate whether multiple physical symptoms are associated with burnout among faculties at a medical university and hospital workers.

## 2. Materials and Methods

### 2.1. Participants

This cross-sectional study was a part of a university survey entitled “A Survey about Raising Children, Caregiving, Work Environment and Satisfaction of Work” which investigated work–life balance of faculties and hospital workers at a large private university in Japan. This university has five campuses that include schools of medicine, pharmacology, medical technology, science and engineering, economics, law, liberal arts, language, and education. At the time of the survey, two-thirds of the faculties belonged to medical-affiliated campuses with three affiliated hospitals; there were 1137 faculties (women: 20%; School of Medicine: 70%) and 2629 hospital workers registered at the three affiliated hospitals (women, 48%). All of these faculties and hospital workers were invited to participate in this study in February 2016. A self-administered questionnaire and informed consent format were sent by post via the administrative office at each campus. The faculties comprised academic personnel—full professors as well as associated and assistant professors—who were employed by the university. A total of 1186 respondents provided consent and completed the self-administered questionnaire (response rate: 31%). We excluded questionnaires with missing values on the 16 physical symptoms (*n* = 42), three types of the Copenhagen Burnout Inventory (CBI) scores (*n* = 20), whether they are faculties or hospital workers (*n* = 14), and gender (*n* = 30). The final sample comprised 1080 participants, including 525 faculties (women: 42%) and 555 hospital workers (women: 88%) who were included in the analyses.

We have followed the ethical standards of the institution and the 1964 Helsinki declaration and its later amendments or ethical standards in conducting this research. This study was approved by the ethics committee in Teikyo University (#TEIRIN 15-141 in 2015).

### 2.2. Measures

#### 2.2.1. Burnout

The primary outcome of this study was the CBI [17,18]. The CBI measures work-related burnout (WBO, 7 items), personal burnout (PBO, 6 items), and client-related burnout (CBO, 6 items). For this study, “clients” refer to students for faculties and to patients for hospital workers. A total of 19 questions were asked. Answers were presented on a 5-point Likert scale ranging from 100 points for “always/to a very high degree” to 0 points for “never/to a very low degree.” If fewer than three questions were answered in the personal and patient-related burnout subscales or fewer than four questions in the work-related burnout subscale, the respondent was classified as a non-responder. The three burnout subscale scores were calculated with one item in reversed order relative to the response; therefore, higher overall scores indicate a higher degree of burnout.

Previous research has demonstrated that all three domains have high internal reliability, particularly among human service workers such as employees at hospitals [19], facilities for people with severe disabilities, and home care services [20]. Existing research also shows that all three burnout domains are associated with increased work absenteeism and predicts intention to leave the workplace [20,21].

#### 2.2.2. Physical Symptoms Perceived

The main exposure variable was the number of self-reported physical symptoms. The 16 physical symptoms included in the questionnaire were headache, eye symptoms, dizziness, nausea, vomiting, diarrhea, constipation, abdominal pain, musculoskeletal pain, articular pain, backache, chest pain, palpitations, dyspnea, insomnia, and fatigue [22,23]. The frequency of these symptoms had 9 possible answers: “almost none”, “less than once a month”, “more than once a month”, “once a week”, “a few times a week”, “4–6 times a week”, “every day”, “more than once a day”, and “always”. We defined physical symptoms as those experienced “once a week or more” (i.e., 1 vs. 0) and summed up the number of symptoms experienced. Based on the median number of 2 (25–75%, 0–4), the total number was grouped into quartiles (i.e., 0/1–2/3–4/5–).

#### 2.2.3. Work–Family Conflict (WFC)

The Work–Family Conflict Scale used in this study was developed by Greenhaus [24] and Carlson [25] and then translated into Japanese [26]. This study uses one question from each of the 6 components of the WFC scale (time-based work interference with family, time-based family interference with work, strain-based work interference with family, strain-based family interference with work, behavior-based work interference with family, behavior-based family interference with work). These questions are as follows: (1) The time I must devote to my job keeps me from participating equally in household responsibilities and activities; (2) The time I spend on family responsibilities often interferes with my work activities; (3) When I get home from work, I am often too frazzled to participate in family activities/responsibilities; (4) Because I am often stressed due to family responsibilities, I have a hard time concentrating on my work; (5) The effective behaviors I perform do not help me to be a better parent and spouse; (6) The behaviors that work for me at home do not seem to be effective at work.

Responses were measured using a 5-point Likert scale from “most agree” (5) to “least agree” (1). The sum of these six items became a continuous variable expressing the degree of WFC.

#### 2.2.4. Work Stress and Social Support

The Job Content Questionnaire (JCQ) [14] is a valid and reliable questionnaire to measure psychological stress and social support in the workplace [27]. Job strain was defined based on the Karasek model, which divides job demand by job control. The Karasek demand–control model is a well-known and established scientific measure for quantifying psychological stress at work. The conceptual idea of job strain is that workplace stress is a function of how demanding a person’s job is and how much control the person has over their responsibilities in terms of discretion, authority, or decision latitude. High strain was defined as >75 percentile and low strain as the ≤75 percentile. Social support was defined as high support (>50 percentile) and low support (≤50 percentile).

#### 2.2.5. Covariates

Other items investigated in this study included age group (i.e., 20s, 30s, 40s, or older), marital status (i.e., married, single, divorced, or widowed), presence of a child or not, housekeeping hours on weekdays (i.e., 1.5 h or more and 1.5 h or less) and weekends (i.e., ≥3 h or <3 h), faculty worker at university or university-affiliated hospital worker, years of work experience, work hours, working nights per month (i.e., ≥3.5 nights or <3.5 nights), and job categories (i.e., medical doctor, pharmacist, nurse, medical engineer, or other).

### 2.3. Statistical Analyses

Variables were statistically investigated to determine if they differed according to the profession group (faculty or hospital worker). Statistical significance was determined using a chi-squared test for categorical variables and a *t*-test or Wilcoxon test for continuous variables. General linear models with the outcome of three subtypes of the CBI subscale scores were applied to investigate whether multiple physical symptoms were associated with the CBI score. Simultaneous regression model was used to determine which variables were significant. Trend P for the linearity of the quartile of the total numbers was estimated in the final multivariable models. Sensitivity analyses were performed to determine how much the following particular symptoms are associated with burnout degree: fatigue, backache, and insomnia. These specific symptoms were chosen as the former two are the most frequent symptoms and insomnia is a risk factor for depression [28]. The statistical interactions between any significant variables were investigated in the final regression models.

The significance level was set at *p* < 0.050. All analyses were performed using SAS software (version 9.4; SAS Institute, Cary, NC, USA).

## 3. Results

### 3.1. Baseline Characterstics and Working Conditions

Table 1 shows the baseline characteristics and working conditions according to profession group. The median number (25–75%) of physical symptoms was 1.0 (0–3) among faculties and 2.0 (1–5) among hospital workers (*p* < 0.001). Of the women, the occupation with the highest percentage was nurses (79%), while among men, most (59%) were doctors. Hospital workers were more likely to be women, single, in their 20s, work longer hours at home on weekdays, and have WFC. The majority of faculties were men, aged ≥40 years, married, and more likely to have a child. Hospital workers perceived higher job strain than faculties did (*p* < 0.001) and had higher burnout scores in all three domains (WBO: 43 vs. 29; PBO: 50 vs. 33; CBO: 33 vs. 29, all *p*-values < 0.001).

### 3.2. Physical Symptoms

Table 2 shows the physical symptoms perceived once per week or more by profession. Of the 16 symptoms, the most frequent complaint was fatigue (55%), followed by backache (37%). These two symptoms as well as insomnia were observed more frequently in hospital workers than in faculties (*p* < 0.001 for fatigue, *p* = 0.023 for backache, and *p* < 0.001 for insomnia).

### 3.3. The Results of the General Linear Models

#### 3.3.1. Factors Associated with WBO, PBO and CBO among Faculties

Table 3 shows the results of the univariable general linear models for factors associated with WBO, PBO, and CBO among faculties. Almost all the covariates were significant except for gender (only CBO), housekeeping hours in weekdays (only CBO), housekeeping hours in weekends (only WBO and CBO), working experiences and working nights per month.

Table 4 shows the multivariable general linear model of WBO, PBO, CBO among faculties. We excluded the working nights per month from covariates because more than half were missing values.

The WBO score increased with the quartile of physical symptoms (+5.5 points for 1–2 symptoms; +8.5 points for 3–4 symptoms; 13 points for 5 or more symptoms, *p* < 0.001), WFC (+1.5 points, *p* < 0.001), and job strain (+12 points, *p* < 0.001). The WBO score (beta estimates) decreased when participants were married (−4.3 score, *p* = 0.027), when they had a child (−4.4 score, *p* = 0.019), and when they had job support (−3.3 score, *p* = 0.016). When either fatigue, backache, or insomnia was entered into the model instead of physical symptom quartile, each symptom was significantly associated with an increased risk of WBO (all *p* values < 0.001), among which fatigue was most strongly correlated with WBO (+8.8 points, *p* < 0.001). The PBO score increased with the quartile of physical symptoms (+9.3 points for 1–2 symptoms; +15 points for 3–4 symptoms; +23 points for 5 or more symptoms, *p* < 0.001), WFC (+1.5 points, *p* < 0.001), job strain (+12 points, *p* < 0.001), and housekeeping hours in weekdays (+4.1 points, *p* = 0.048). The PBO score decreased when participants had workplace support (−4.5 points, *p* = 0.004). When either fatigue, backache, or insomnia was entered into the model instead of physical symptom quartile, each symptom was significantly associated with an increased risk of PBO (all *p* values < 0.001), among which fatigue was most strongly correlated with PBO (+15 points, *p* < 0.001).

The CBO score increased with WFC (+1.3 points, *p* < 0.001), and job strain (+9.6 points, *p* < 0.001). The CBO score (beta estimates) decreased when participants were married (−4.8 score, *p* = 0.031). The quartile of physical symptoms, backache and insomnia was insignificant while fatigue became significant (*p* = 0.043).

#### 3.3.2. Factors Associated with WBO, PBO and CBO among Hospital Workers

Table 5 shows the results of the univariable general linear models for factors associated with WBO, PBO, and CBO among hospital workers. All the covariates were significant except for housekeeping hours in weekdays and weekends (only PBO), gender and housekeeping hours in weekdays (only CBO).

Table 6 shows multivariable general linear models for factors associated with WBO, PBO and CBO among hospital workers. The WBO score increased with the quartile of physical symptoms (+7.9 points for 1–2 symptoms; +11 points for 3–4 symptoms; 18 points for 5 or more symptoms; *p* < 0.001), WFC (+1.7 points, *p* < 0.001), and job strain (+4.6 points, *p* = 0.003). The WBO decreased when the participants had a child (−7.6 points, *p* = 0.001) and job support (−4.7 points, *p* = 0.001). When either fatigue, backache, or insomnia was entered into the model instead of physical symptom quartile, each symptom was significantly associated with an increased risk of WBO (all *p* values < 0.001), among which fatigue was most strongly correlated with WBO (+13 points, *p* < 0.001).

The PBO score increased with the quartile of physical symptoms (+14 points for 1–2 symptoms; +18 points for 3–4 symptoms; +28 points for 5 or more symptoms; *p* < 0.001), younger age group (+7.4 points in 20s, and +4.4 in 30s, *p* = 0.032), WFC (+1.6 points, *p* < 0.001), work hours (+1.2 points per one hour increase, *p* = 0.025), and job strain (+7.3 points, *p* < 0.001). The PBO score decreased when they had a child (−5.3 points, *p* = 0.040). When either fatigue, backache, or insomnia was entered into the model instead of physical symptom quartile, each symptom was significantly associated with an increased risk of PBO (all *p* values < 0.001), among which fatigue was most strongly correlated with PBO (+20 points, *p* < 0.001).

The CBO score increased with quartile of physical symptoms (+6.8 points for 1–2 symptoms; +3.8 points for 3–4 symptoms; 13 points for 5 or more symptoms, *p* < 0.001), WFC (+1.7 points, *p* < 0.001), more working nights per month (+4.1 points, *p* = 0.019) while the score decreased when the participant had a child (−7.7 points, *p* = 0.004). When either fatigue, backache, or insomnia was entered into the model instead of physical symptom quartile, each symptom was significantly associated with an increased risk of CBO (*p* values = 0.002, <0.001, and 0.003, respectively), among which backache was most strongly correlated with CBO (+7.0 points, *p* < 0.001).

## 4. Discussion

This study found that the perception of more physical symptoms was significantly and linearly associated with higher burnout scores in all three Copenhagen burnout subscales, with one exception: CBO among faculties. Overall, job strain and WFC were associated with an increased risk of burnout, which may increase intention to leave their profession, as suggested by previous studies [29,30]. In addition, we demonstrated that being married (WBO and CBO among faculties), having a child (except for PBO and CBO among faculties), and job support (faculty and hospital workers with WBO and faculties with PBO) were associated with a decreased risk of burnout, suggesting that social support from family members and colleagues at work may mitigate burnout levels.

In this study, we focused on the number of physical symptoms, rather than the individual symptoms themselves. Previous studies have demonstrated that psychological stress has a marked impact on physical symptoms, including fatigue [31], bowel changes (diarrhea and constipation) [32], insomnia [33], and chronic pain or discomfort [34]. It sometimes results in a life-threatening disease, such as depression and its attendant risk of suicide or self-harm [35] or coronary heart disease [36]. Indeed, our study confirmed that the most frequent symptoms—fatigue and backache, as well as insomnia—were independently associated with burnout, except for CBO among faculties. Furthermore, it should be noted that the beta estimate of fatigue was almost equivalent to those of three or four physical symptoms in all of the CBI subscales. This result may be interpreted to mean that multiple physical symptoms could be useful to identify burnout if there are more than five physical symptoms. Indeed, referring to the Diagnostic and Statistical Manual of Mental Disorders, Fifth Edition, fatigue is a symptom of major depression; therefore, it is well known to be highly correlated with psychological distress, including burnout. Our study results are unique in that they show that a greater number of perceived physical symptoms, rather than fewer symptoms, was associated with higher degrees (i.e., beta estimates) of burnout.

It is often difficult for an employee to report a mental health problem to their employer, as reporting it may directly cause job termination. The most frequently reported mistreatment in the workplace due to reporting job stress is an unwanted lay-off, followed by fewer opportunities for promotion and salary increase. In Japan, the Labor Contract Act prohibits employers from forcing the termination of an employee for medical reasons, such as depression, without considering the possibility of recovery; however, it still occurs [37]. In these circumstances when maltreatment/discrimination occurs, workers do not consult with the industrial doctors directly hired by the hospitals they work for because they believe that those industrial doctors are linked with the employer. The Japan Labor Safety and Hygiene Law introduced a mental health screening program (i.e., Act 66) at periodic health checkups in 2015; however, some workers are not willing to take the tests and tell their employer the truth. The situation is more serious among medical professionals because they understand that reporting a mental health problem to their employer may create additional disadvantages. In this regard, the results of the present study are useful, as physical symptoms might serve as a warning of an employee’s mental health status.

Among the three types of burnout, our multivariable regression model demonstrated that physical symptoms were not associated with CBO among faculties. In our study, burnout scores of the three domains were smaller among faculties than among hospital workers. The discrepancy in burnout levels between the two professions may be due to labor characteristics. Most hospital workers in our participant group are nurses, who are primarily responsible for the psychological and emotional aspects of patients associated with illnesses, while faculties take care of students rather than patients. The discrepancy may be explained by the “client” (patients for hospital workers and students for faculties) each professional takes care of, with patients being more demanding in clinical settings than students.

Inconsistent results have been reported in the past on the effect of having a child as a mitigating or risk factor for burnout. Due to gender roles in traditional Japanese society and the traditional division of labor augmenting WFC among high-achieving professionals, having a child has long been believed to cause career obstacles among professional women [38]. However, having a child functioned as a mitigating factor of burnout in our study, despite the presence of “work–life conflict” [39]. When children are in early childhood, parents are more likely to commit to child-rearing, which might affect WFC. Although our dataset does not include the age of participants’ children, considering the participants’ average age (38 years), their children might be grown enough to be more independent, which may have a buffering effect on the correlation between having children and burnout. Previously, Sarason et al. [40]. indicated that perceived social support can affect an individual’s emotional well-being. A previous study similarly demonstrated that social support factors from “family” and “friends” have consistently shown the strongest associations with symptomatology among clinically distressed people and a student sample [41]. The significant and independent effect of having a child, apart from marital status and other social support, suggests the possible buffering effect of having a child on burnout.

Our study had several limitations. First, the response rate for our survey was low, at 31%, which may result in nonresponse bias. Second, because this study was embedded in a periodic university survey about work–life balance conducted by a women’s support center, women were more likely to participate from both source populations of faculties (women’s response rate 88%: women constitute 20% of 1137 faculty) and hospital workers (women’s response rate 42%: women constitute 48% of 2629 healthcare professionals). In contrast, those who face greater work demands (and may be at higher risk of burnout as a result) may have opted out, which might lead to an underestimation. Third, our study population was derived from the faculties of one private university, indicating that generalizability may be limited to some extent. Fourth, unmeasured confounding factors may still have existed, even if we had made an effort to include personal factors along with working conditions to explain burnout.

Although our results require careful interpretation due to these limitations, we believe that our study is valid in its conclusion that a greater number of physical symptoms perceived could indicate burnout. The practical implication may include the various physical symptoms in the periodical health check-up questionnaire. Those who manifested five or more physical symptoms perceived more often than once a week may be at risk of burnout. Thus, the number of physical symptoms may be a useful indicator to identify a high-risk individual who may require early intervention.

## 5. Conclusions

Individuals with more physical symptoms had higher burnout scores on all three CBI subscales, except for CBO among faculties, even after adjusting for potential confounding variables. As employees may find physical symptoms easier or more socially acceptable to report than mental health issues, it is feasible to count the number of symptoms and incorporate them into the periodic health checkup questionnaire. This may help to identify individuals at risk of burnout. The number of physical symptoms may become an easy, simple, and useful indicator to identify high-risk individuals in the workplace, particularly those in the medical field, who are more likely to experience burnout.

## Figures and Tables

**Table 1 ijerph-18-03246-t001:** Baseline characteristics and working conditions according to profession.

	Total *n* = 1080 Percent (%)	Faculty *n* = 555 Percent (%)	Hospital Worker *n* = 525 Percent (%)	*p*-Value
Gender				<0.001
Women	65	42	88	
Men	35	58	12	
Age group				<0.001
20s	35	17	52	
30s	22	21	23	
40s or older	43	62	25	
Marital status				<0.001
Married	50	66	35	
Single	50	34	65	
Presence of a child				<0.001
Yes	43	55	33	
No	57	45	67	
Housekeeping hours in weekdays, median (25%, 75%)	1.5 (1.0, 3.0)	1.0 (0.5, 2.0)	2.0 (1.0, 3.0)	<0.001
Housekeeping hours in weekend, median (25%, 75%)	3.0 (2.0, 5.0)	3.0 (1.5, 4.0)	3.0 (2.0, 5.0)	0.330
Work Family Conflict, median (25%, 75%)	17 (14, 19)	16 (13, 18)	18 (15, 20)	<0.001
Working Experience years, median (25%, 75%)	9.0 (3.0, 19)	11 (5.0, 22)	6.0 (3.0, 15)	<0.001
Work hours, median (25%, 75%)	9.0 (8.0, 10)	9.0 (8.0, 11)	9.0 (8.0, 10)	<0.001
Working nights per month, median (25%, 75%)	3.0 (0, 5.0)	2.0 (0, 4.0)	3.0 (0, 5.0)	0.002
Job strain				<0.001
High strain	25	15	33	
Low strain	75	85	67	
Support				0.510
High support	53	54	52	
Low support	47	46	48	
Physical symptom median, (25%, 75%)	2.0 (0, 4.0)	1.0 (0, 3.0)	2.0 (1.0, 5.0)	<0.001
Burnout, median (25%, 75%)				
Work-related	36 (21, 50)	29 (18, 43)	43 (24, 54)	<0.001
Personal	42 (25, 58)	33 (21,50)	50 (33, 67)	<0.001
Client-related	33 (21, 46)	29 (21, 42)	33 (21, 50)	<0.001
Job category				<0.001
Doctor	19	23	4.0	
Pharmacist	3.1	1.0	3.6	
Nurse	65	21	6	
Engineer	13	3.2	17	
Other	25	48	0	

**Table 2 ijerph-18-03246-t002:** Physical symptoms perceived once in a week or more by profession.

	Total (*n* = 1080)	Faculty (*n* = 555)	Hospital Worker (*n* = 525)	*p*-Value
	%	%	%	
Fatigue	55	49	61	<0.001
Backache	37	33	40	0.023
Eye symptom	24	24	25	0.690
Headache	21	16	25	<0.001
Constipation	20	16	23	<0.001
Insomnia	17	14	21	<0.001
Diarrhea	12	11	13	0.300
Abdominal pain	12	7.6	16	<0.001
Tinnitus	11	10	11	0.510
Arthritis	10	17	9.2	0.190
Myalgia	9.4	9.1	9.7	0.740
Shortness of breath	9.1	9.1	9.1	0.100
Palpitation	7.9	6.9	8.8	<0.001
Dizziness	7.1	5.3	8.8	0.030
Nausea	6.9	4.6	9.2	<0.001
Chest pain	2.7	3.6	1.8	0.070

**Table 3 ijerph-18-03246-t003:** Univariable models of an effect of covariates according to burnout subscale among faculties.

		WBO			PBO			CBO		
		β	SE	*p*-Value	β	SE	*p*-Value	β	SE	*p*-Value
Women vs. men		8.6	1.6	<0.001	12	1.9	<0.001	1.5	1.6	0.343
Age group vs. 40s or older				<0.001 d			<0.001 d			0.007 d
	20s	13	2.1		14	2.6		4.8	2.1	
	30s	5.1	2.0		5.5	2.4		4.3	1.9	
Marital status (Married vs. Single)		−13	1.6	<0.001	−13	1.9	<0.001	−7.4	1.6	<0.001
Presence of a child		−11	1.6	<0.001	−11	1.9	<0.001	−6.4	1.6	<0.001
Housekeeping hours in weekdays		4.0	1.6	0.014	7.7	1.9	<0.001	0.41	1.5	0.790
Housekeeping hours in weekend		2.0	1.6	0.219	5.0	1.9	0.009	0.60	1.5	0.693
WFC		2.2	0.17	<0.001	2.5	0.21	<0.001	1.8	0.17	<0.001
Working experience (years)		−3.2	1.7	0.053	−1.6	2.0	0.415	−2.1	1.6	0.191
Work hours		1.3	0.39	0.001	1.4	0.46	0.002	1.3	0.37	<0.001
Working nights per month		4.1	2.2	0.061	4.1	2.7	0.134	2.9	2.2	0.183
High strain vs. Low strain		23	2	<0.001	26	2.4	<0.001	15	2.0	<0.001
High support vs. Low support		−8.0	1.6	<0.001	−11	1.8	<0.001	−5.1	1.5	<0.001
Physical symptoms (Reference: none)				<0.001 d			<0.001 d			<0.001 d
	1–2	8.8	1.8		13	2.1		6.7	1.9	
	3–4	15	2.1		22	2.3		7.3	2.1	
	5−	21	2.2		31	2.4		9.1	2.2	
Individual models for three particular symptoms										
	Fatigue	16	1.4	<0.001	23	1.6	<0.001	8.2	1.5	<0.001
	Backache	12	1.6	<0.001	16	1.9	<0.001	4.3	1.6	0.008
	Insomnia	15	2.2	<0.001	20	2.6	<0.001	7.2	2.2	0.001
d trend P										

WBO: work-related burnout PBO: personal burnout CBO: client-related burnout WFC: work–family conflict.

**Table 4 ijerph-18-03246-t004:** Multivariable general linear models of an effect of covariates according to burnout subscale among faculties.

		WBO			PBO			CBO		
		(*n* = 462, R2 49%)			(*n* = 462, R2 51%)			(*n* = 462, R2 26%)		
		β	SE	*p*-Value	β	SE	*p*-Value	β	SE	*p*-Value
Women vs. Men		0.180	1.6	0.855	2.2	1.9	0.258	3.3	1.9	0.072
Age group vs. 40s or older				0.171			0.072			0.532
	20s	4.0	2.6		6.2	3.1		−2.6	2.9	
	30s	0.28	1.8		0.39	2.1		−0.04	2.0	
Marital status (Married vs. Single)		−4.3	1.9	0.027	−3.0	2.3	0.186	−4.8	2.2	0.031
Presence of a child		−4.4	1.8	0.019	−3.5	2.1	0.106	−3.2	2.1	0.132
Housekeeping hours in weekdays		1.3	1.8	0.514	4.1	2.1	0.048	−1.2	2.0	0.562
Housekeeping hours in weekend		−2.4	1.7	0.170	−2.6	2.0	0.190	0.48	2.0	0.809
WFC		1.5	0.17	<0.001	1.5	0.19	<0.001	1.3	0.19	<0.001
Working experience		−0.45	1.7	0.920	0.77	2	0.703	−1.9	2.0	0.335
Work hours		0.33	0.33	0.271	0.41	0.39	0.287	0.48	0.38	0.205
High strain vs. Low strain		12	2.0	<0.001	12	2.3	<0.001	9.6	2.2	<0.001
High support vs. Low support		−3.3	1.3	0.016	−4.5	1.6	0.004	−2.2	1.5	0.145
Physical symptoms (Reference: none)	0	−		<0.001 d	−		<0.001 d			0.176 d
	1–2	5.5	1.6		9.3	1.8		3.8	1.8	
	3–4	8.5	1.9		15	2.2		2.7	2.1	
	5–	13	2		23	2.4		3.1	2.3	
Individual models for three particular symptoms										
	Fatigue a	8.8	1.4	<0.001	15	1.6	<0.001	3.2	1.6	0.043
	Backache b	7.0	1.5	<0.001	9.5	1.8	<0.001	0.85	1.60	0.603
	Insomnia c	7.5	2.0	<0.001	12	2.4	<0.001	1.8	2.2	0.411
WBO: work-related burnout PBO: personal burnout CBO: client-related burnout WFC: work–family conflict										
a *n* = 462 R248%, *n* = 462 R250%, *n* = 462 R2 26%										
b *n* = 462 R246%, *n* = 462 R243%, *n* = 462 R2 25%										
c *n* = 462 R245%, *n* = 462 R243%, *n* = 462 R2 25%										
d trend P										

**Table 5 ijerph-18-03246-t005:** Univariable models of an effect of covariates according to burnout subscale among hospital workers.

		WBO			PBO			CBO		
		β	SE	*p*-Value	β	SE	*p*-Value	β	SE	*p*-Value
Women vs. men		6.9	2.6	0.008	6.4	2.9	0.029	3.9	2.6	0.138
Age group vs. 40s or older				<0.001 d			<0.001 d			<0.001 d
	20s	13	2.0		12	2.1		8.4	2.1	
	30s	6.4	2.4		6.9	2.7		3.8	2.5	
Marital status (Married vs. Single)		−11	1.7	<0.001	−11	2.0	<0.001	−6.7	1.8	<0.001
Presence of a child		−12	2	<0.001	−11	2.1	<0.001	−9.1	1.8	<0.001
Housekeeping hours in weekdays		−3.5	1.8	0.048	−3.0	2.0	0.133	−3.3	1.8	0.062
Housekeeping hours in weekend		−3.8	1.7	0.028	−2.7	2.0	0.169	−4.3	1.7	0.014
WFC		2.2	0.2	<0.001	2.4	0.21	<0.001	1.9	0.19	<0.001
Working experience (years)		−9.2	1.7	<0.001	−8.7	1.9	<0.001	−5.1	1.7	0.004
Work hours		1.9	0.56	<0.001	2.8	0.63	<0.001	2.0	1.6	<0.001
Working nights per month		6.4	1.7	<0.001	5.7	1.9	0.003	8.2	1.7	<0.001
High strain vs. Low strain		13	1.7	<0.001	16	1.9	<0.001	10	1.8	<0.001
High support vs. Low support		−8.7	1.7	<0.001	−8.5	1.9	<0.001	−5.4	1.7	0.002
Physical symptoms (Reference: none)				<0.001 d			<0.001 d			<0.001 d
	1–2	9.2	2.2		15	2.4		7.1	2.3	
	3–4	15	2.3		21	2.6		6.9	2.5	
	5–	25	2.3		35	2.5		18	2.4	
Individual models for three particular symptoms										
	Fatigue	19	1.5	<0.001	26	1.6	<0.001	11	1.7	<0.001
	Backache	11	1.7	<0.001	16	1.9	<0.001	11	1.7	<0.001
	Insomnia	16	2.0	<0.001	17	2.3	<0.001	9.1	2.1	<0.001
d trend P										

WBO: work-related burnout PBO: personal burnout CBO: client-related burnout WFC: work–family conflict.

**Table 6 ijerph-18-03246-t006:** Multivariable general linear models of an effect of covariates according to burnout subscale among hospital workers.

		WBO			PBO			CBO		
		(*n* = 466, R2 47%)			(*n* = 466, R2 49%)			(*n* = 466, R2 32%)		
		β	SE	*p*-Value	β	SE	*p*-Value	β	SE	*p*-Value
Women vs. Men		2.9	2.2	0.195	1.5	2.5	0.545	2.1	2.5	0.415
Age group vs. 40s or older				0.297 d			0.032 d			0.764 d
	20s	8.2	3.4		7.4	3.9		3.2	4	
	30s	3.4	2.1		4.4	2.3		−1.0	2.4	
Marital status (Married vs. Single)		−6.5	2.1	0.435	−2.9	2.3	0.205	1.3	2.4	0.593
Presence of a child		−7.6	2.3	0.001	−5.3	2.6	0.040	−7.7	2.6	0.004
Housekeeping hours in weekdays		−0.93	1.8	0.611	−0.87	2.1	0.674	−1.2	2.1	0.555
Housekeeping hours in weekend		−0.41	1.8	0.823	1.0	2.0	0.634	−1.6	2.1	0.447
WFC		1.7	0.17	<0.001	1.6	0.19	<0.001	1.7	0.20	<0.001
Working experience		1.7	3.0	0.567	0.41	3.4	0.905	3.2	3.5	0.354
Work hours		0.27	0.48	0.574	1.2	0.54	0.025	0.42	0.55	0.444
Working nights per month		0.70	1.5	0.645	−0.63	1.7	0.711	4.1	1.7	0.019
High strain vs. Low strain		4.6	1.5	0.003	7.3	1.7	<0.001	3.1	1.8	0.079
High support vs. Low support		−4.7	1.4	0.001	−2.7	1.6	0.097	−2.6	1.7	0.117
Physical symptoms (Reference: none)				<0.001 d			<0.001 d			<0.001 d
	1–2	7.9	2.0		14	2.2		6.8	2.3	
	3–4	11	2.2		18	2.5		3.8	2.5	
	5–	18	2.2		28	2.4		13	2.5	
Individual models for three particular symptoms										
	Fatigue a	13	1.5	<0.001	20	1.7	<0.001	5.6	1.8	0.002
	Backache b	6.0	1.5	<0.001	10	1.8	<0.001	7.0	1.7	<0.001
	Insomnia c	11	1.8	<0.001	12	2.1	<0.001	6.1	2.0	0.003
WBO: work-related burnout PBO: personal burnout CBO: client-related burnout WFC: work–family conflict										
a *n* = 466 R2 48%, *n* = 466 R2 50%, *n* = 466 R2 29%										
b *n* = 466 R2 41%, *n* = 466 R2 38%, *n* = 466 R2 30%										
c *n* = 466 R2 43%, *n* = 466 R2 39%, *n* = 466 R2 29%										
d trend P										

## Data Availability

Our data are available upon reasonable request.

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
