# Peer review of "Multiple Physical Symptoms Are Useful to Identify High Risk Individuals for Burnout: A Study on Faculties and Hospital Workers in Japan"

_ijerph, 2021, doi:10.3390/ijerph18063246_

Round 1
Reviewer 1 Report
This manuscript is in good shape. My only recommendations for improving it are:
- Better describe in the abstract the effect of marital status.
- Abandon the stepwise regression and replace it with a simultaneous regression.

Author Response
#Reviewer 1
This manuscript is in good shape. My only recommendations for improving it are:
- Better describe in the abstract the effect of marital status.
⇒Thank you for your suggestion. We changed “Marital status” to “Being married” in the abstract.
- Abandon the stepwise regression and replace it with a simultaneous regression.
⇒In this revision, we simultaneously entered all variables into the logistic regression models. Accordingly, we have minor changes in Tables but the main results remained same.
Reviewer 2 Report
A very interesting study, but it requires modification before proceeding further: The introduction is very short. The constructs and concepts necessary to understand the manuscript are not explained. The sample size estimated based on which statistic formula? What are the exclusion criteria? For best comparison of two groups, the matching in some confounders is necessary. Both groups differ substantially with respect to core sociodemographic variables; analysis needs to account for this (include them as control variables). Occupational load varies depending on the requirements at the workplace (e.g. work in the emergency department vs. screening point). Did the authors include such analysis in the study? I propose to broaden the "Discussion" section to make it more valuable, using the publications from recent years. Separate a section of limitations and implications for professional practice. Adapt references to editorial requirements.
Author Response
Reviewer 2
A very interesting study, but it requires modification before proceeding further:
⇒Thank you for your comments, let us answer your questions as below.
#1The introduction is very short. The constructs and concepts necessary to understand the manuscript are not explained.
⇒We included concepts necessary to understand the research hypothesis that there might be associations between physical symptoms and burnout scores.
p.1, line 34-37
Usually, they claim various kinds of physical symptoms but healthcare professionals often cannot find any underlying diseases. In such cases, we had the hypothesis that it can be due to mental health malfunction. There are a few studies to support our theory and one previous study suggested that patients with multiple unspecified physical complaints often have mental health problems [1,2].
#2The sample size estimated based on which statistic formula?
We used sample size calculation for simple linear regression with 4 predictors, α=0.05, power=0.8, and then found that a sample of 85 will identify the model with R2=0.13. (or f=0.3873 or f2=0.15). In this study, we had a much larger sample size of 1,080: 525 faculties and 555 hospital workers.
#3What are the exclusion criteria?
⇒Exclusion criteria are written on page 2, lines 87-90.
We excluded questionnaires with missing values on the 16 physical symptoms (n = 42), three types of the Copenhagen Burnout Inventory (CBI) scores (n = 20), whether they are faculties or hospital workers (n = 14), and gender (n = 30). (p2. 87-90)
#4For best comparison of two groups, the matching in some confounders is necessary. Both groups differ substantially with respect to core sociodemographic variables; analysis needs to account for this (include them as control variables).
⇒We completely agree with the reviewer’s opinion. Accordingly, we first divided into two groups (faculties vs. hospital workers) and then built multivariable linear regression models with adjustment of confounders.
#5Occupational load varies depending on the requirements at the workplace (e.g. work in the emergency department vs. screening point). Did the authors include such analysis in the study?
⇒We thank the reviewer for the productive comment. We did not ask the department occupational load in this study. Because participants included nurses, physicians, pharmacists, and faculties, not all the participants belonged to a particular department. Nevertheless, we measured workload by using JCQ scores instead. It is previously known that JCQ scores indicate the stress level of individuals at the workplace. We adjusted the job stress in multivariable logistic regression models.
#6I propose to broaden the "Discussion" section to make it more valuable, using the publications from recent years.
⇒In our study, we applied classic concepts of “Job Content Questionnaire”, and “Work-Family Conflict” and thus the related references are from the old days. For “Burnout”, the concept is relatively new and thus its related reference is published in the recent years of 2007 (#14 in references). Otherwise, we cited relatively new articles published in the late 2010s including #16 and #18 in 2016, #39 in 2017, and #29 in 2019.
#7 Separate a section of limitations and implications for professional practice.
⇒Thank you for your suggestion. We separated a section of the limitations and implications from this study according to your advice (p.11, 392-394) and added some suggestions about how to use physical symptoms for the screening of burnout (p.11, 394-p.12, 398).
p.11, line 392 - p.12, line 398
Although our results require careful interpretation due to these limitations, we believe that our study is valid in its conclusion that a greater number of physical symptoms perceived could indicate burnout. The practical implication may include the various physical symptoms in the periodical health check-up questionnaire. Those who manifested five or more numbers of physical symptoms perceived more often than once a week may be at risk of burnout. Thus, the number of physical symptoms may be a useful indicator to identify a high-risk individual who may require early intervention.
#8 Adapt references to editorial requirements.
Referring to the author's instruction, we updated references.
Reviewer 3 Report
This study was carried out about five years ago and “ask why did you take so long time before your article submitted.
Table 1 and 2 are easy to understand however they could be improved in clarity by only keeping the percentages and not the numbers in the columns. The total number is in the top so its clear to see whre the percentage comes from.
For all tables, 1-4 please take care (probably the publisher) so the tables should not be divided between two .
In the tables please also consider if needed to keep the explanation for the abbreviations like CBI WBO in the tables should be explained.
For the mat and methods section please explain how more in details in practice the questionnaires were distributed and answered,
it looks like it was sent out by email and then returned self administrated from the participants phone! Which programe is used, how long time did it take to comply and how many reminders were send out?
When you recommend multiple physical symptoms might be useful for high risk persons for burnout, which standardised scheme do you recommend?
Thank you for this excellent study
Author Response
Reviewer 3
Comments and Suggestions for Authors
#1 This study was carried out about five years ago and “ask why did you take so long time before your article submitted.
⇒The first author works as a full-time anesthesiologist and has to work on the article in a weekend. She is also an adult graduate student and has a lot of lectures to take. It took 5 years because she could not take enough time to finish it sooner, however, we believe that her distinguished background gave her various points of view as a clinical physician, a mother of a child, and a researcher.
#2 Tables 1 and 2 are easy to understand however they could be improved in clarity by only keeping the percentages and not the numbers in the columns. The total number is at the top so it is clear to see where the percentage comes from.
⇒Thank you for your suggestion, we revised Tables 1 and 2.
#3 For all tables, 1-4 please take care (probably the publisher) so the tables should not be divided between two.
⇒Thank you for your comment, we totally agree with you for readability. We updated Table 2-4 and added extra 2 more tables in this revision for clarity of data presentation.
#4 In the tables please also consider if needed to keep the explanation for the abbreviations like CBI WBO in the tables should be explained.
⇒Thank you for your suggestion, we added abbreviations list in a footnote.
#5 For the mat and methods section please explain how more in detail in practice the questionnaires were distributed and answered, it looks like it was sent out by email and then returned self-administrated from the participants' phone! Which program is used, how long time did it take to comply and how many reminders were sent out?
⇒Thank you for asking. We recruited our sample by post in February 2016 and simultaneously sent a self-administered questionnaire and informed consent format. We did not send a reminder and the time span of the investigation was one month. We updated 2.1. Participants in Materials and Methods.
p2. Line 79-84
At the time of the survey, two-thirds of the faculties belonged to medical-affiliated campuses with three affiliated hospitals; there were 1,137 faculties (women: 20%; School of Medicine: 70%) and 2,629 hospital workers registered at the three affiliated hospitals (women, 48%). All of these faculties and hospital workers were invited to participate in this study in February 2016. A self-administered questionnaire and informed consent format were sent by post via the administrative office at each campus.
#6 When you recommend multiple physical symptoms that might be useful for high-risk persons for burnout, which standardized scheme do you recommend?
⇒Thank you for your suggestion.
We recommend including the various physical symptoms in the periodical health check-up. Those who demonstrated 5 or more physical symptoms perceived more often than once a week might be at risk of burnout.
We updated our conclusion at p.11, line 392-p.12, line 398.
Although our results require careful interpretation due to these limitations, we believe that our study is valid in its conclusion that a greater number of physical symptoms perceived could indicate burnout. The practical implication may include the various physical symptoms in the periodical health check-up questionnaire. Those who manifested five or more numbers of physical symptoms perceived more often than once a week may be at risk of burnout. Thus, the number of physical symptoms may be a useful indicator to identify a high-risk individual who may require early intervention.
Thank you for this excellent study
⇒I appreciate your comments. Thank you very much for your review.
Reviewer 4 Report
Thank you for an excellent study using validated questionnaires, a large sample and a well designed study. The statistical analysis is very rigorous and on the whole the manuscript is well-written and well presented. However, I found the two tables presenting the outcome of the stepwise multiple regression impossible to follow. I think there is too much rather than to little information, For this kind of analysis I am more used to seeing a list of variables with standardised beta values (which you do mention in the discussion) - to judge the relative influence of the variables - and p-values to indicate the significant ones. Is it possible that these tables could be simplified in the main text and, if possible, the existing tables appended as supplementary material?
Author Response
Reviewer 4
Thank you for an excellent study using validated questionnaires, a large sample, and a well-designed study. The statistical analysis is very rigorous and on the whole, the manuscript is well-written and well presented. However, I found the two tables presenting the outcome of the stepwise multiple regression impossible to follow. I think there is too much rather than to little information, For this kind of analysis I am more used to seeing a list of variables with standardized beta values (which you do mention in the discussion) - to judge the relative influence of the variables - and p-values to indicate the significant ones. Is it possible that these tables could be simplified in the main text and, if possible, the existing tables appended as supplementary material?
⇒ Thank you for your suggestion, we tried to modify these tables as simply as possible. For readability, we divided univariable from multivariable tables so the numbers of tables increased from 4 to 6 in this revision. Following another reviewer’s suggestion, we changed the method of analyses from stepwise to simultaneous entry, so there are minor changes of results emphasized in yellow. We really appreciate your warm comments.
Round 2
Reviewer 2 Report
Accept in present form